
**Landslide susceptibility assessment based on different machine-learning**
**methods in Zhaoping County of eastern Guangxi**
Chunfang Kong [1, 2, 3, 4], Kai Xu [1, 2, 3, *], Junzuo Wang [1], Chonglong Wu [1, 2, 3], Gang Liu [1, 2]
[1]School of Computer, China University of Geosciences, Wuhan, 430074, China
[2]Hubei Key Laboratory of Intelligent Geo-Information Processing, Wuhan, 430074, China
[3]Innovation Center of Mineral Resources Exploration Engineering Technology in Bedrock Area,
Ministry of Natural Resources, Guiyang, 550081, China
[4]National-Local Joint Engineering Laboratory on Digital Preservation and Innovative
Technologies for the Culture of Traditional Villages and Towns, Hengyang, 421000, China
Corresponding author: Kai Xu
E-mail address: xukai@cug.edu.cn
Phone: +862767883286, +8615327232692
Mobile: +862767883051





**Abstract:** Regarding the ever increasing and frequent occurrence of serious landslide disaster in
eastern Guangxi, the current study were implemented to adopt support vector machines (SVM),
particle swarm optimization support vector machines (PSO-SVM), random forest (RF), and
particle swarm optimization random forest (PSO-RF) methods to assess landslide susceptibility by
Zhaoping County. To this end, 10 landslide disaster-related causal variables including digital
elevation model (DEM)-derived, meteorology-derived, Landsat8-derived, geology-derived, and
human activities factors were selected for running four machine-learning (ML) methods, and
landslide susceptibility evaluation maps were produced. Then, receiver operating characteristics
(ROC) curves, and field investigation were performed to verify the efficiency of these models.
Analysis and comparison of the results denoted that all four ML models performed well for the
landslide susceptibility evaluation as indicated by the values of ROC curves -- from 0.863 to 0.934.
Moreover, the results also indicated that the PSO algorithm has a good effect on SVM and FR
models. In addition, such a result also revealed that the PSO-RF and PSO-SVM models have the
strong robustness and stable performance, and those two models are promising methods that could
be transferred to other regions for landslide susceptibility evaluation.
**Keywords:** Landslide; Susceptibility evaluation; Machine-learning (ML); Particle swarm
optimization (PSO); Support Vector Machines (SVM); Random Forest (RF)



## 1. Introduction

The geological environment in eastern Guangxi is fragile and landslide disaster occur

frequently, which not only causes huge economic losses and ecological damage, but also seriously

restricts the survival of human beings and the sustainable development of human society

(Pourghasemi et al., 2012; Huang and Zhao, 2018; Chen et al., 2019). With the rapid development

of the economy in recent decades, the frequency and intensity of landslide disaster are rapidly

increasing with the over-exploitation and utilization of natural resources by humans (Zhang et al.,

2016). Therefore, it is of great significance to objectively evaluate the landslide susceptibility for

the reduction and prevention of the disasters.

In recent years, more and more machine-learning (ML) algorithms have been optimized and

applied for landslide susceptibility assessment in different regions. Examples are: Bayesian

network (BN) (Song et al., 2012; Pham et al., 2016), Naïve Bayes (NB) (Tien Bui et al., 2012;

Pham et al. 2015, 2016), artificial neural networks (ANN) (Choi et al., 2012; Zare et al., 2013;

Conforti et al., 2014; Pham et al. 2015; Xu et al., 2015; Tien Bui et al., 2016; Aditian et al., 2018;

zhou et al., 2018), Support Vector Machines (SVM) (Marjanović et al., 2011; Tien Bui et al.,

2012; 2016; Pourghasemi et al., 2013; Pradhan, 2013; San, 2014; Kavzoglu et al., 2014; Peng et

al., 2014; Hong et al. 2015; Pham et al., 2016; Kumar et al., 2017; Ada and San, 2018; zhou et al.,

2018; Aktas and San, 2019; Wang et al., 2019; Zhang et al., 2019), Logistic Regression (LR)

(Choi et al., 2012; Kavzoglu et al., 2014; Hong et al. 2015; Trigila et al., 2015; Pham et al., 2016;

Tien Bui et al., 2016; Lin et al., 2017; Sevgen et al., 2019; Wang et al., 2019), decision tree (DT)

(Tien Tien Bui et al., 2012; Pradhan, 2013; Tsai et al., 2013; Youssef et al., 2016; Hong al., 2018;





Khosravi et al., 2018; Aktas and San, 2019), Random Forest (RF) (Trigila et al., 2015; Youssef et
al., 2016; Chen et al., 2017; Ada and San, 2018; Aktas and San, 2019), Fisher's linear
discriminant analysis (FLDA) (Rossi et al., 2010; Murillo-García and Alcántara-Ayala, 2015),
SVM-ANN (Xia et al., 2018), SVM-LR (Wang et al., 2019), convolutional neural network
(CNN)-SVM, CNN-RF and CNN-LR (Fang et al., 2020). These have all been used to
quantitatively predict and assess the susceptibility for landslide in different regions of the world.
These studies play an important role in the susceptibility evaluation and prediction of landslide.

In addition, many comparative studies on landslide susceptibility assessment using different

ML methods have been performed. For example, Marjanović et al. (2011) stated a comparison
research of SVM with other models and found that SVM has the best performances compared with
DT and LR for landslide susceptibility evaluation. In another landslide assessment investigation,
Tien Bui et al. (2012) also proved that the capability of SVM was better than the decision tree and
NB models. Another comparative investigation, Trigila et al. (2015) completed a comparison of the
LR and RF algorithms in an analytic study of landslide susceptibility and discovered that RF
presents a better performance than LR. Another comparative study on performance of landslide
susceptibility mapping, Kavzoglu et al. (2014) made an experimental research to investigate that
the performance of SVM is higher than the LR. Another study certified that results produced from
SVM have the highest prediction accuracy compared to LR, BN, NB, and FLDA for landslide
susceptibility evaluation (Pham et al., 2016). Likewise, another comparative research on the
performance of two ML algorithms, SVM and FR, for landslide susceptibility prediction based on
two-level random sampling, was compared by Ada and San (2018).



In general, each of the above ML models has been widely applied to landslide prediction and
evaluation. Among them, SVM and RF have been widely proved to be useful methods in the
evaluation of landslide susceptibility (Marjanović et al., 2011; Tien Bui et al., 2012; Kavzoglu et
al., 2014; Trigila et al., 2015; Pham et al., 2016; Ada and San, 2018). However, few studies have
focused on the optimization of SVM and RF models in landslide susceptibility prediction and
evaluation and compared the optimized results. Therefore, the objective of the present paper is to:
(1) determine the landslide susceptibility assessment factors by multi-source data fusion and
correlation factor analysis; (2) optimize SVM and RF models by using a particle swarm
optimization (PSO) algorithm; (3) analyze and evaluate the susceptibility levels of landslide by
using the SVM, PSO-SVM, RF, and PSO-RF models for Zhaoping County; and (4) compare the
performances of four ML models for landslide susceptibility evaluation by receiver operating
characteristic (ROC) curve, statistic analysis, and field-verified methods. The results provide
valuable informational support for the prediction and evaluation of landslide in Zhaoping County,
Guangxi.

## 2. Study areas and materials

### 2.1. Study areas

Zhaoping County is located between longitude 110°34′E to 111°19′E and latitude 23 ′39'N to 24 ′24'N in the eastern part of Guangxi, the middle reaches of the Guijiang River, with a total area of about 3,223.67km$^2$ and a total population of 448,000, as shown in Fig. 1. It is situated in the subtropical monsoon humid climate region with mild climate and abundant rainfall.  The annual average temperature is 19.8 ℃ and the annual rainfall is 2046 mm, which is one of the rainy and heavy rain centers in Guangxi.

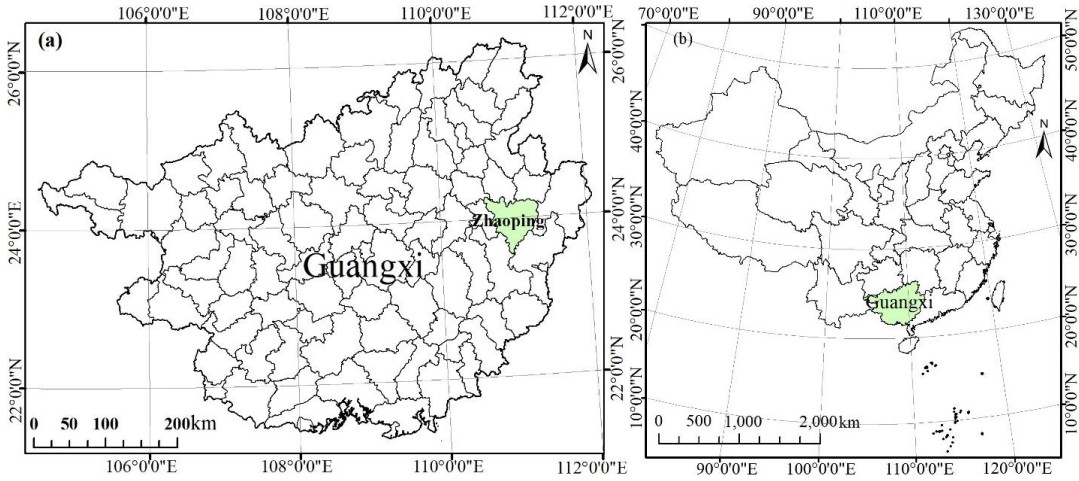

**Fig. 1.** Location of Zhaoping County in Guangxi Province (a) and China (b)

Zhaoping County has remarkable geomorphological characteristics; it is in a mountainous region with intervening deep valleys, where the mountain area is 87.6% of the total area, and the terrain is high in the northwest and low in the southeast. The main structure is near EN to WS trending large fault and north protruding Dayaoshan arc structural compression belt, where a





series of secondary arc folds and faults are distributed. At the same time, the Dayaoshan uplift
belt is cut by a series of near-SN trending faults and it forms many secondary depression areas.
Under the influence of multi-stage tectonic movements, joint fissure is developed in rock mass
and rock is weathered seriously, which provides the basic conditions for the formation of
landslide. Finally, extremely fragile geological characteristics are formed, because of long-term
geological changes in geological internal and external forces; these landslide occured frequently
in Zhaoping County. According to the detailed survey data of landslide in 2018 in the Guangxi
Geological Survey Bureau, there are 345 hidden danger points of  landslide in Zhaoping County.
**2.2. Data sources and hazards inventory data**
Following are the main data sources adopted in this paper: (1)  A digital elevation model
(DEM) for Zhaoping County with a spatial resolution of 30m×30m; it was constructed from
ASTER Global DEM acquired from the United States Geological Survey
(http://earthexplorer.usgs.gov). Based on the DEM data, three geomorphic factors were generated:
slope, aspect, and plan curvature. (2) The annual precipitation data was collected from the
government of Guangxi Meteorological Bureau; (3) Landsat 8 OLI image (2017/12/24, 124/043)
used to extract the normalized differential vegetation index (NDVI), and land use and land cover
(LULC) map; (4) 1:50 000 topographic map was collected to reflect the densities of residents and
road network. (5) 1:50 000 geological map was adopted to extract the stratum lithology and
tectonic complexity. (6) A landslide inventory map in Zhaoping County was prepared from field
investigation of Guangxi Geological Survey Bureau.


## 2.3. Classification of evaluation factors


There are many kinds of factors affecting the occurrence of landslide in Zhaoping County,
and the factors are not independent of each other. To more objectively assess the susceptibility of
landslide, a total of ten hazards affecting factors were chosen based on the results of field
investigation of Guangxi Geological Survey Bureau and the characteristics of landslide
distribution in Zhaoping County; they are slope, aspect, curvature, annual rainfall, NDVI, stratum
lithology, tectonic complexity, LULC, residential density, and road network density. At the same
time, these factors have been classified into different grades (Table 1) according to the analysis of
influence of each evaluation factor to landslide occurrences implemented by Guangxi Geological
Survey Bureau for Zhaoping County.

**Table 1** Landslide affecting factors and their classes

| No. | Evaluation factor | Classification |
|---|---|---|
| (a) | Slope (°) | 1-[0,7); 2-[7,13); 3-[13,19); 4-[19,25); 5-[25,34); 6-[34,50); 7-[50,70); 8-[70,76) |
| (b) | Aspect (°) | 1-[0,22.5); 2-[22.5,67.5); 3-[67.5,112.5); 4-[112.5,157.5); 5-[157.5,202.5); 6-[205.2,247.5); 7-[247.5,292.5); 8-[292.5,360) |
| (c) | Plan curvature | 1-[-25,-5); 2-[-5,-2.5); 3-[-2.5,-1); 4-[-1,0); 5-[0,1); 6-[1,2.5); 7-[2.5,5); 8-[5,28.9] |
| (d) | Annual rainfall (mm) | 1-[0,1980); 2-[1980,2100); 3-[2100,2220); 4-[2220,2340); 5-[2340,2460); 6-[2460,2580); 7-[2580,2700); 8-[2700,2820] |
| (e) | NDVI | 1-[0,0.01); 2-[0.01,0.09); 3-[0.09,0.17); 4-[0.17,0.25); 5-[0.25,0.33); 6-[0.33,0.4); 7-[0.4,0.5); 8-[0.5,0.71] |
| (f) | Stratum lithology | 0-River; 1-Quaternary; 2-carbonate rock; 5-clasolite intercalated with siliceous rocks; 6-clastic rock; 7-sandstone and shale; 8-granite or basal rocks |
| (g) | Tectonic complexity | 1-[0,1.4); 2-[1.4,2.7); 3-[2.7,3.8); 4-[3.8,4.9); 5-[4.9,6); 6-[6,7.3); 7-[7.3,8.9); 8-[8.9,9.4] |
| (h) | LULC | 1-cultivated land; 2-woodland; 3-grassland; 4-river and lake; 5-construction land |
| (i) | Residential density | 1-[0,1.2); 2-[1.2,2.7); 3-[2.7,4.5); 4-[4.5,6.9); 5-[6.9,10.1); 6-[10.1,14.2); 7-[14.2,19.7); 8-[19.7,25] |
| (j) | Road network density (km/km²) | 1-[0,3.2); 2-[3.2,4.7); 3-[4.7,6.1); 4-[6.1,7.8); 5-[7.8,9.7); 6-[9.7,11.7); 7-[11.7,13.9); 8-[13.9,14] |

According to the classification standard of Table 1, the attribute value of each evaluation



factor is obtained by superimposed analysis with a 30m*30m grid and the attributes of each
evaluation factor; the results are shown in Fig. 2(a-j). Thereinto, Fig. 2(a-c) indicates that maps of
slope (Fig. 2a), aspect (Fig. 2b), and curvature (Fig. 2c) were extracted from DEM with a
30m*30m grid cell, which represented the influence of topography on the development and
distribution of landslide in Zhaoping County.

Precipitation, especially heavy rain or continuous precipitation is the external dynamic

factor that induces landslide (Zhang et al., 2016). There is plenty of precipitation in Zhaoping
County, and the annual average number of heavy rain days is between 3 and 15 days. Under the
action of precipitation infiltration, scour, erosion, and so on, unstable mountains easily form
landslide. Meanwhile, the landslide and frequent periods of heavy rain are basically the same,
both concentrated from May to August, indicating that the formation of landslide is closely
related to heavy rain in Zhaoping County. Figure 2d is the annual rainfall map of Zhaoping
County from the Guangxi Meteorological Bureau.

The ecological environment is closely related to the occurrence of landslide. Zhaoping

County has a warm and humid climate with a wide variety of vegetation. In this current study, the
map of NDVI (Fig. 2e) was extracted from a Landsat8 OLI image to characterize the ecological
environmental characteristics for Zhaoping County.

The strata of Zhaoping County are mainly Cambrian, Devonian, and a small number of

Quaternary, and the main lithology  are clastic rocks, clastic rocks intercalated with siliceous
rocks, sandstone and shale, carbonate rock, and a small amount of granite or basal rock,
accounting for 55.89%, 34.11%, 4.54%, 3.96%, and 0.47% of the total area, respectively (Fig. 2f).





Clastic rocks are prone to landslides under the action of precipitation, especially heavy
precipitation (Zhang et al., 2016). At the same time, after the influence of multi-stage tectonic
movement and long-term action of geological internal and external forces, a more complex
geological structure pattern is formed, and folds and fractures staggered distribution, which
resulted in extremely fragile geological environmental characteristics in Zhaoping County. Figure
2g indicates the tectonic complexity of Zhaoping County.
In addition, human activities have become one of the major driving forces for environmental
changes and induced landslide (Zhang et al., 2016). Human engineering activities such as land
use change, steep slope reclamation, road and bridge building, development of forests and
mineral resources, construction of hydropower engineering and so on, strongly disturb the
topography and geomorphology and make it lose its equilibrium state, which leads to the
probability of landslide occurring far more than in the natural state.  Therefore, LULC map,
residential density, and road network density were selected as representative factors to reflect the
influences of human activities on the environment in Zhaoping County, as shown in Fig. 2(h-j).



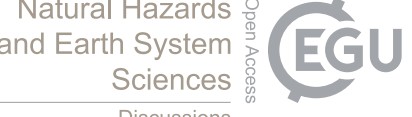







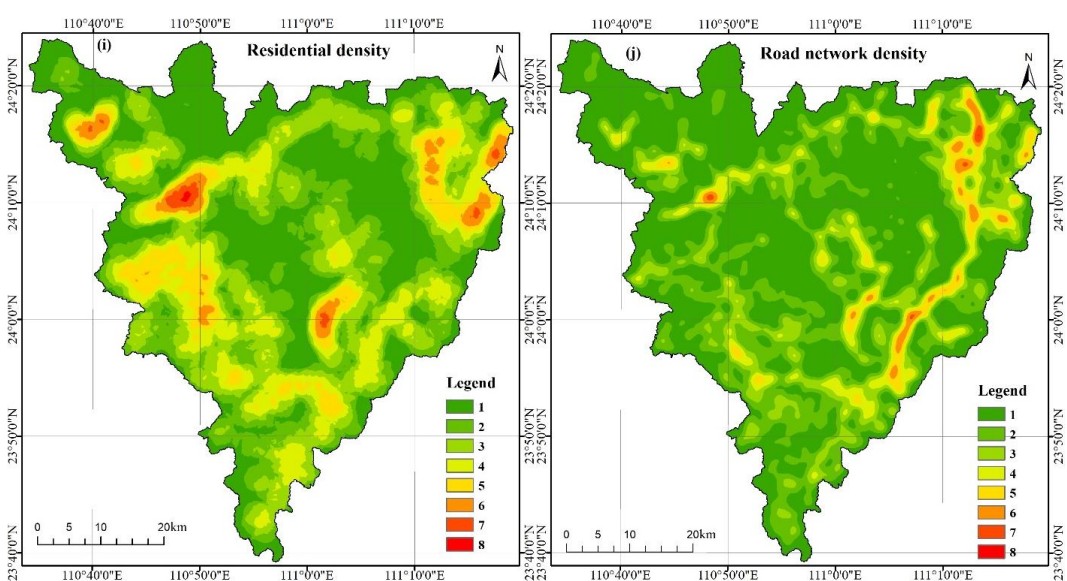

**Fig. 2.** Attribute value of landslide evaluation factors [(a) slope, (b) Aspect, (c) Plan curvature, (d) Annual

rainfall, (e) NDVI, (f) Stratum lithology, (g) Tectonic complexity, (h) LULC, (i) Residential density, (j) Road

network density]

On the basis of the above, the database of the landslide susceptibility evaluation factors in

Zhaoping County was established, with a total of 3,581,859 grid evaluation units. In the present

database, 1,493 grid units as training samples were selected to construct the training dataset,

including 242 landslide hazards points and 1,251 non-hazards points; 1,042 grid units as testing

samples to construct the testing dataset, including 103 landslide hazards points and 939

non-hazards points. Four ML models (SVM、PSO-SVM、RF and PSO-RF) for geological hazard

susceptibility evaluation were trained using the training dataset, whereas the performance of the

constructed four landslide susceptibility evaluation models was verified using the testing dataset.



## 3. Methods

Landslide susceptibility evaluation has been carried out in nine main processes (Fig. 3): (1)

According to the environmental characteristics of Zhaoping County, all the evaluation factors

related to landslide are collected; (2) Evaluation units were divided into 30m×30m grid cells by

using ArcGIS; (3) The landslide susceptibility assessment factor system was determined; (4)

Classification criterion for each evaluation factor was divided according to the classification

standard of Guangxi Geological Survey Bureau; (5) Spatial and attribute databases for each

evaluation factor were set up based on 30m*30m grid cells; (6) Training and testing datasets were

selected; (7) Landslide susceptibility evaluation models were established based on different ML

methods, such as SVM, PSO-SVM, RF, and PSO-RF; (8) We validated and compared the

evaluation accuracy for four ML models with ROC curves, statistical analysis, and field-survey;

And (9) we divided the landslide susceptibility levels in Zhaoping County.

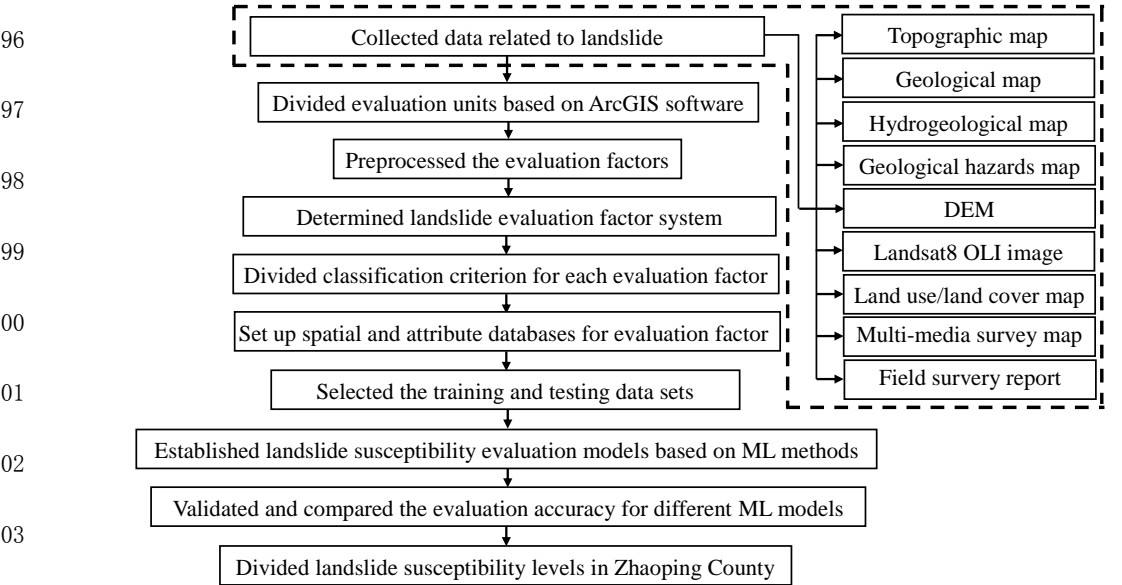

**Fig. 3.** Flowchart of landslide susceptibility evaluation based on ML





**3.1. SVM model**
Support vector machine (SVM) is based on statistical approach and structured risk
minimization theory (Cortes and Vapnik, 1995; Vapnik, 1995). It uses kernel function to map the
input variables to a high-dimensional characteristic space, and then finds the optimal hyperplane
for separating two classes. The SVM ensures that the extreme solution is the global optimal
solution (Kavzoglu et al., 2014). At present, SVM has been proven to have many unique
advantages in dealing with small samples, nonlinear and high-dimensional pattern recognition, and
is successfully applied in hazards prediction and assessment ( Marjanović et al., 2011; Tien Bui et
al., 2012; Pradhan, 2013; Kavzoglu et al., 2014; Pham et al., 2016; Ada and San, 2018).
In the landslide assessment of the current study, the training sample set is given as $\{x_i, y_i\}, i =$
$1, 2, \dots, n; \; x_i \in R^m, \; y_i \in \{-1, +1\}$. SVM seeks the optimal classification superplane in the
feature space of the landslide, which can separate the two types of training samples of the hazards
point and the non-hazards point. The optimal classification superplane is defined as the following:
$$\min_{w,b} \frac{1}{2}\|w\|^2 \qquad (1)$$
$$\text{s.t.} \; y_i(w^T x_i + b) \geq 1, i = 1, 2, \dots\dots, m$$

where $n$ represents the number of training samples, $m$ represents the dimension of the input
vector, $\|w\|$ represents the norm of the superplane normal vector, and $b$ is the displacement term.
The Lagrangian multiplier rule is introduced to find the extreme value, and the auxiliary
function is generated as follows:
$$\text{L}(w, b, \lambda) = \frac{1}{2}\|w\|^2 - \sum_{i=1}^{m} \lambda_i \left( y_i(w^T x_i + b) - 1 \right) \qquad (2)$$

where the $\lambda_i$ is Lagrange multiplier.





The dual minimum method given by Vapnik (1995) and Tax and Duin (1999) is used to solve
the $w$ and $b$ values of the equation.
For the nonlinear non-separable hazards samples, the non-negative relaxation variables ($\xi_i$)
and penalty factor $C$ are introduced to adjust the constraint conditions, and the formula is modified
to:
$$\min_{w,b} \frac{1}{2}\|w\|^2 + C\sum_{i=1}^{m}\xi_i$$
$$\text{s.t. } y_i(w^T x_i + b) \geq 1 - \xi_i, i = 1, 2, \ldots\ldots, m \qquad (3)$$

where $\xi_i > 0$ denotes a sample classification error; $C$ represents the degree of the penalty. In the
landslide assessment, $C \in (0,1]$ denotes that the support vector represents the percentage of the
entire training set. Therefore, the smaller the valve of $C\sum_{i=1}^{n}\xi_i$, the better for finding the
classification hyperplane.
Meanwhile, the radial basis kernel function $k(x, x_i)$ is adopted to process the nonlinear
decision boundary when the SVM is constructed based on the training sample set. As shown in the
formula (4):
$$k(x, x_i) = \exp\left(-\frac{\|x - x_i\|^2}{2\sigma^2}\right) \qquad (4)$$

where $\sigma^2$ represents the kernel parameter, which implicitly decides the distribution of data after
mapping to a new characteristic space. The number of support vectors affects the speed of
training and prediction.
To bring the kernel function into (3), the final regression function (the optimal hyperplane) is
obtained as formula (5):
$$g(x) = \sum_{i=1}^{n} \lambda_i y_i k(x_i, x) + b \qquad (5)$$



The evaluation results of landslide susceptibility in Zhaoping County are obtained by using
regression analysis of formula (5) and parameter optimization. Furthermore, the natural breakpoint
method is adopted to divide the susceptibility into five levels: extremely high, high, middle, low,
and extremely low areas (Fig. 4a).
**3.2. SVM model based on particle swarm optimization (PSO-SVM)**
From the above analysis, it can be seen that the selection of the SVM parameters (penalty
factor $C$, and the core parameter of radial basis function $\sigma$) directly affects the prediction
accuracy of the landslide susceptibility evaluation model (Kavzoglu et al., 2014). Therefore, the
particle swarm optimization (PSO) algorithm with powerful parameter global search capability
was adopted to select the optimal $C$ and $\sigma$, and the PSO-SVM model for prediction and evaluation
of landslide was set up in Zhaoping County. The main steps of the PSO-SVM model can be
summed up as Table 2:
**Table 2** The main steps of the PSO-SVM model

**(1) Initialization:**

The initial parameters of the PSO-SVM model are set, including species size, iteration times, learning factor, inertia weight, initial particle and particle initial velocity. The particle vector represents a SVM model corresponding to different $C$ and $\sigma$.

**(2) Optimization:**

In the process of particle optimization, each solution of the optimization problem is called a particle in the search space. The particle adaptation value ($f_i$) is calculated according to the fitness function. Adaptive function is the measure basis of the selection individual, and the individual is evaluated by the fitness function.

**(3) Replacement:**

On the basis of the objective function, the adaptive value of each particle ($f_i$), the population individual optimal solution $f_i(p_{best})$, and the population global optimal solution $f_i(p_{gbest})$ were calculated and compared. If $f_i <$ $f_i(p_{best})$, then the optimization solution of the previous round is replaceed with the new adaptation value ($f_i$), and the particles of the previous round is replaced with the new particles, and then the $f_i(p_{best})$ of each particle is compareed with the $f_i(p_{gbest})$ of all particles. If $f_i(p_{best}) < f_i(p_{gbest})$, the optimal solution of each particle is used to replace the optimal solution of all the original particles, and the current state of the particles is saved at the same





| | |
|---|---|
| time. | |
| **(4) Determination:** | |
| If the $f_i$ of the individual in the population meets the requirements, or if the evolutionary algebra is terminated, then the calculation is ended, and the particle individual corresponds to the optimal $C$ and $\sigma$ combination, otherwise go to step (2) to continue the iteration. | |
| **(5) Set up the PSO-SVM model:** | |
| The global optimal PSO-SVM model is obtained by using the optimal parameters of the SVM with the optimal $C$ and $\sigma$ combination to train the training samples. The susceptibility of landslide is quantitatively evaluated and divided into five levels: extremely high, high, middle, low, and extremely low areas (Fig. 4b). | |

### 3.3. Random Forests (RF) model

Random Forests (RF) is a cluster tree classification proposed by Breiman (2001), which is composed of several unrelated decision trees. It is put back from the original training dataset by the Bagging algorithm to obtain multi-Bootstrap training data sets. And then the corresponding decision tree model was acquired by training random selection of $m$ attributes from all $M$ decision attributes. Finally, the final classification result of the test set samples was determined by voting.

Suppose that for the landslide sample $x$ of Zhaoping County, the output of the $g$ decision tree is $f_{tree,g}(x) = i, i = 1,2,…,n$, that is, its corresponding category, $g = 1,2,…,G$, G is the number of decision trees in RF, then the output of the RF model is as follows:

$$f_{RF}(x) = \underset{i=1,2,…,n}{\arg} \max\{G(f_{tree,g}(x) = i)\} \qquad (6)$$

where $G(\cdot)$ represents the number of samples that satisfy the expressions in parentheses.

The construction process of the RF model for landslide susceptibility assessment in Zhaoping County is as Table 3:

**Table 3** The main steps of the RF model

| |
|---|
| **(1) Initialization:** |





Suppose *D* is an original training data set of landslide susceptibility assessment factors, which is composed of *M* prediction attributes (*M*=10) and a classification attribute *Y* (*Y* =5). There is *n* (n=3,581,859 different examples in *D*.

**(2) Get multiple training datasets:**

The *K* new training subsets of {D$_1$, D$_2$, …, D$_K$} were obtained by *K* times random sampling with replay from the original training data set *D* by using Bagging algorithm. At the same time, each of the *K* training subsets contains *n* instances, in which there is repetition.

**(3) Training to generate decision tree:**

For each training subset $D_i$ (1≤i≤K), the decision tree without pruning is generated by the following procedure:

Firstly, let the number of predictive attributes in the training sample be *M*, *F* (F<M) attributes are randomly chosen from *M* to compose a random characteristic subspace X$_i$, and those as the split attribute sets of the present node of the decision tree. In the process of generating the RF model, the value of *F* remains unaltered;

Secondly, the node was split according to the optimal split attribute of each node selecting from the random feature subspace X$_i$ by the decision tree generation algorithm;

Thirdly, every tree grows completely and has no pruning process. The corresponding decision tree h$_i$(D$_i$) is generated by each training set D$_i$;

Fourthly, the FR model of {h$_1$(D$_1$), h$_2$(D$_2$), …, h$_i$(D$_i$)} was generated by combining all the generated decision trees. And the corresponding classification result of {C$_1$(X), C$_2$(X), …, C$_K$(X)} is obtained by using testing of each decision tree h$_i$(D$_i$) with test set sample X;

Finally, according to the classification results of K decision trees, the final classification results corresponding to the test set sample X was determined by classification results with large number of decision trees by voting method.

**(4) Dividing levels:**

According to the above steps, the landslide susceptibility of Zhaoping County is divided into 5 levels (Fig. 4c).

## 3.4. Weighted random forest based on particle swarm optimization algorithm (PSO-RF)

In order to further compare the performance of different models in the evaluation of the susceptibility of the landslide, the parameters of the weighted FR are optimized by the PSO algorithm, and the main steps are as Table 4:

**Table 4** The main steps of the PSO-FR model

**(1) Initialization:**

The initial parameters of the PSO-FR model are set, including number of decision tree *R*, pruning threshold *ε*, number of predicted test samples *X*, and initial value of random attributes *m*.

**(2) Sampling:**





Using rhe Bootstrap algorithm, R training sets are randomly produced, and X pre-test samples are selected in each training set.

**(3) Generating decision tree:**

A total of *R* decision trees is generated by using the rest of the samples of each training set. In the process of generating decision trees, *m* attributes are selected from all attributes as the decision attributes of the present node before each attribute is selected.

**(4) Determination:**

When the number of samples included in the node is less than the threshold ε, the node is taken as the leaf node, and the mode of the target attributes is returned as the classification result of the decision tree.

**(5) Setting up the PSO-SVM model:**

When all decision trees are produced, each decision tree is pre-tested and its weights are calculated by using the following formula:

$$w_r = \frac{X_{correct,r}}{X}, r = 1,2,\dots,R \qquad (7)$$

where $X_{correct,r}$ is the classified correct number of samples of *r* decision trees, and *X* is the number of pre-tested samples.

**(6) Calculation of the classification results:**

The classification results of the model are calculated by the following formula:

$$\int_{WRF}(x) = \underset{i=1,2,\dots,c}{\arg\ max} \left\{ \sum_{r \in R, \int_{tree,r}(x)=i} w_r \right\} \qquad (8)$$

**(7) Optimization:**

Taking the classification results as the fitness values, the PSO algorithm is applied to optimize the parameters of formula (6) iteratively and determined the parameters of the final RF model.

**(8) Running**

Finally, the optimized parameters are input into the model, and the output results of the model are obtained. According to the results, the susceptibility of landslide is divided into five levels (Fig 4d).



## 4. Results and discussions

### 4.1. Evaluation results

The 3,581,859 grids of Zhaoping County were input into the aboved four ML models, and

homologous output values were obtained. According to these output results, the landslide

susceptibility of Zhaoping County was divided into five levels: very low, low, moderate, high and

very high, as shown in Fig. 4.

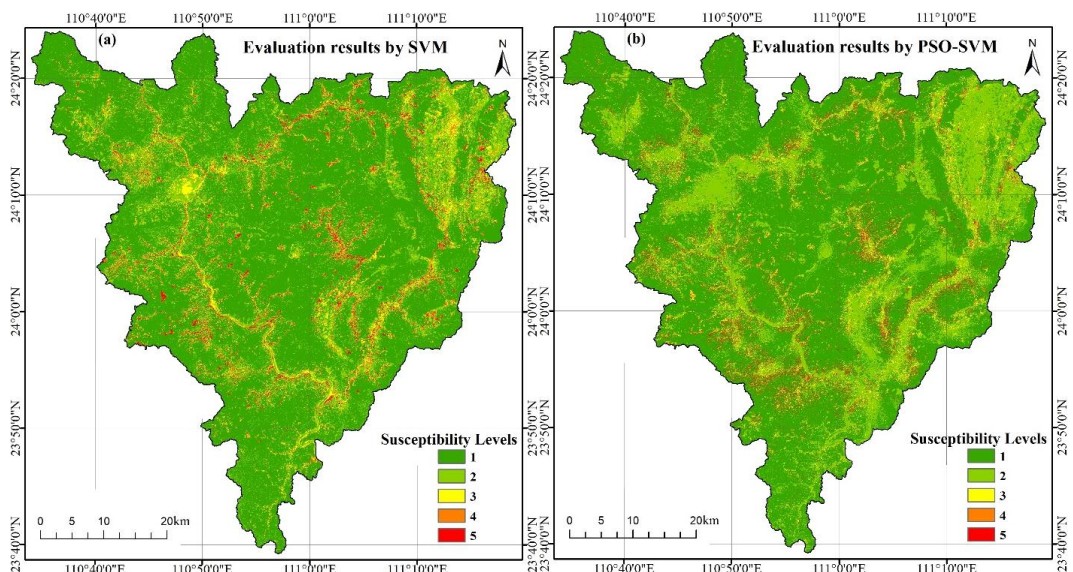

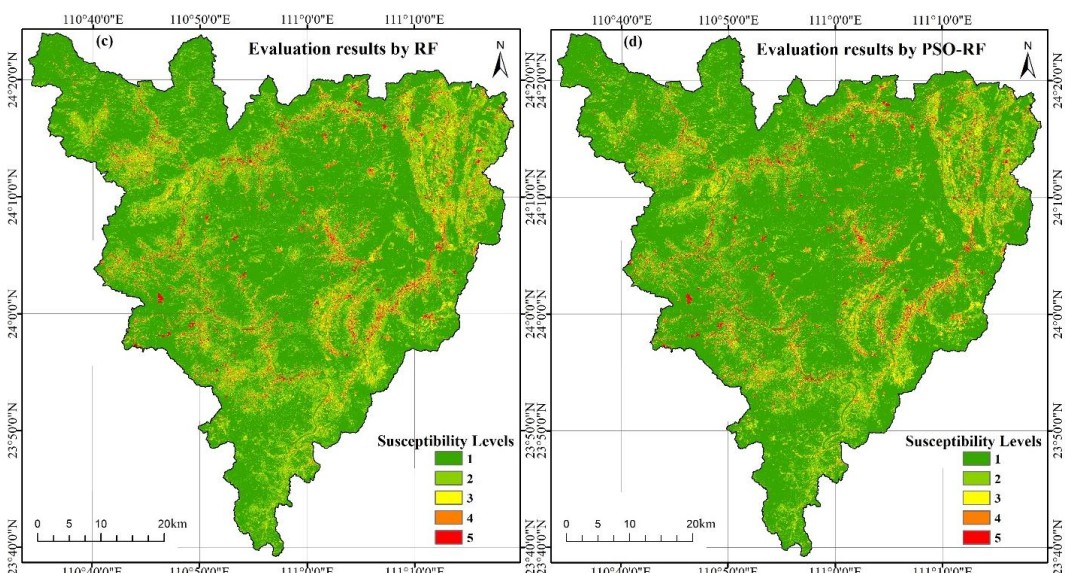

**Fig. 4.** Evaluation results of landslide susceptibility for four ML models in Zhaoping County [1-extremely low,

2-low, 3-middle, 4-high, 5-extremely high; (a) SVM; (b) PSO-SVM; (c) RF; (d) PSO-RF]

Figure 4 shows that the extremely high susceptibility levels for landslide is mainly

distributed in the clastic rock areas along the Guijiang River and its tributaries, and the closer the

river bank, the higher its susceptibility index. Here the geological structure is complex, where

multi-period tectonic movement makes the joints and fractures of rock mass develop, the

weathering of rock is serious, and water erosion is strong. Under the action of precipitation,

especially heavy precipitation, as well as undermining and erosion of river water, clastic rocks are

easy to form landslide disaster.

Simultaneously, Fig. 4 indicates that the high susceptibility levels for landslide is mainly

distributed in the surrounding towns and trunk lines built near the mountains or the Guijiang

River. Here the geological structure is relatively complex, the stability of rock is poor and

weathering is strong, which supplies adequate material basis for the development of landslide



disaster. Simultaneously, the NDVI map of these regions indicate that the vegetation coverage in
these regions is low, which indirectly reflects the frequent human engineering activities in the
region, indicating that the human engineering construction strongly interferes with the geological
ecological environment of the region and leads to the frequent occurrence of landslide. This also
illustrates that the stability and bearing capacity of regional geological environment system
should be fully considered in the construction of human engineering.
Figure 4 also indicates that the medium susceptibility levels for landslide is mainly
distributed along the county roads, rural roads and residential areas, distributed in belts or
surface-like distribution. The rock mass here is stable; the vegetation covers well, and is less
disturbed by human activities.
The remaining areas are low and extremely low susceptibility levels for landslide, far away
from the Guijiang River and its tributaries, with high vegetation coverage and less human
engineering activities.
**4.2. Evaluation accuracy and validation analysis**
Evaluation accuracy and validation analysis is an essential component in landslide
susceptibility prediction and evaluation to attest the availability and scientific significance of the
adopted method (Frattini et al., 2010). Many research papers confirmed that the area under curve
(AUC) of the receiver operating characteristic (ROC) curve was an effective method for the
precision inspection of the prediction model, and was widely used in all subjects (Hanley and Mc
Neil, 1983; Fawcett, 2005; Rossi et al., 2010; Pham et al., 2016; Tien Bui et al.,2016; Chen et al.,


2017; Lin et al., 2017; Hong et al., 2018; Ciurleo et al., 2019). Therefore, the AUC values of the
ROC curves were used to evaluate the accuracy of landslide susceptibility in Zhaoping County
for the ML methods, such as the SVM, PSO-SVM, RF, and PSO-RF model, as shown in Fig. 5.

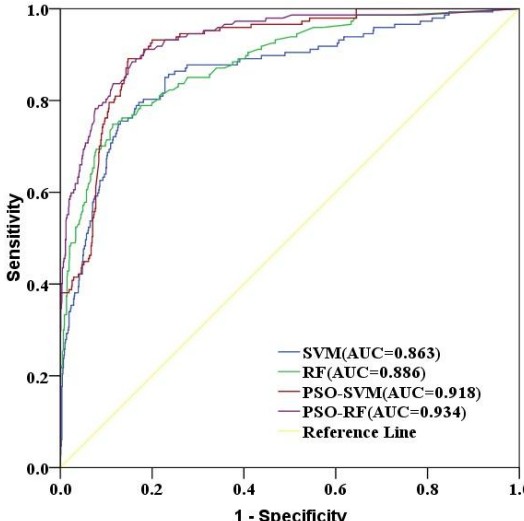


**Fig. 5.** ROC curves and AUC values of validation set for the PSO-RF, RF, PSO-SVM, and SVM model

Figure 5 indicates the ROC curves and the AUC values of the validation set for the PSO-RF,

RF, PSO-SVM, and SVM models. The values of AUC are 0.934, 0.886, 0.918, 0.863,
respectively, which indicate that the accuracy of the four ML methods in the evaluation and
prediction of landslide susceptibility in Zhaoping County is higher than 86%. At the same time,
the AUC values of the PSO-SVM and PSO-RF  models (0.918 and 0.934) were higher than those
of the traditional SVM and the RF (0.863 and 0.886), which indicated that the PSO algorithm can
effectively optimize SVM and RF models, and the prediction accuracy of the optimized model is
more than 91.5%. Such a result further revealed that the PSO-RF and PSO-SVM models have the
stronger robustness and stable performance. Furthermore, the present study further testified that



PSO has strong global parameter search ability, and parameter adjustment is simple and easy to
implement, which confirmed that the PSO algorithm is successfully applied in landslide hazards
evaluation and prediction (Liu et al., 2012; Feng et al., 2017; Hoang and Tien Bui, 2018).

Figure 5 indicates that the performance of the RF and RF-PSO is better than the SVM and

PSO-SVM in evaluating the susceptibility of landslide because the values of AUC for RF (0.886)
and RF-PSO (0.934) are higher than the values of AUC for SVM (0.863) and PSO-SVM (0.918),
respectively, which confirmed that the generalization performance of the integrated learner is
superior to that of a single learner (Li et al., 2014; Zhang et al., 2018). At the same time, the
research further certified that the RF and PSO-RF models have advantages in dealing with high
dimensional features and geological big data, such as fast classification speed, strong anti-noise
ability, and avoiding over-fitting (Tien Bui et al., 2016). However, because of the sensitivity of
the RF and PSO-RF models to the proportion of landslide samples, it is necessary to carry out
sample screening before using RF and PSO-RF models to evaluate the susceptibility of landslide.

In order to further verify the accuracy of the four ML models, the ratio of grid number of

landslide points that fall into different susceptibility levels was counted, as shown in Table 5:
**Table 5** Percentages of landslide points falling into different susceptibility levels

| Susceptibility levels | SVM (%) | PSO-SVM (%) | RF (%) | PSO-RF (%) |
|---|---|---|---|---|
| Extremely high | 0.1238 | 0.2030 | 0.1793 | 0.2306 |
| High | 0.0561 | 0.0609 | 0.0596 | 0.0845 |
| Medium | 0.0302 | 0.0232 | 0.0171 | 0.0117 |
| Low | 0.0124 | 0.0057 | 0.0077 | 0.0041 |
| Extremely low | 0.0010 | 0.0006 | 0.0008 | 0.0005 |

Table 5 indicates that the proportions of hazards points falling into  extremely high and high

susceptibility regions are 0.2306% and 0.0845%, 0.2030% and 0.0609%, 0.1793% and 0.0596%,


and 0.1238% and 0.0561% for the PSO-FR, PSO-SVM, RF, and SVM models, respectively,
which certified that the evaluation accuracy of four ML models in the extremely high and high
prone regions from high to low are: PSO-RF, PSO-SVM, RF, and SVM. Simultaneously, Table 5
also indicates that the proportions of landslide points falling into low and extremely low
susceptibility regions are 0.0041% and 0.0005%, 0.0057% and 0.0006%, 0.0077% and 0.0008%,
and 0.0124% and 0.0010% for the PSO-FR, PSO-SVM, RF, and SVM models, respectively,
which certified that the wrong accuracy of four ML models in the low and extremely low
susceptibility regions from low to high are: PSO-RF, PSO-SVM, RF, and SVM.

In addition to the above two methods of verification, field investigation has been

implemented by Guangxi Geological Survey Bureau in Zhaoping County. Simultaneously, the
field investigation results were compared and analyzed with the evaluation results of four ML
models, as shown in Fig. 6:

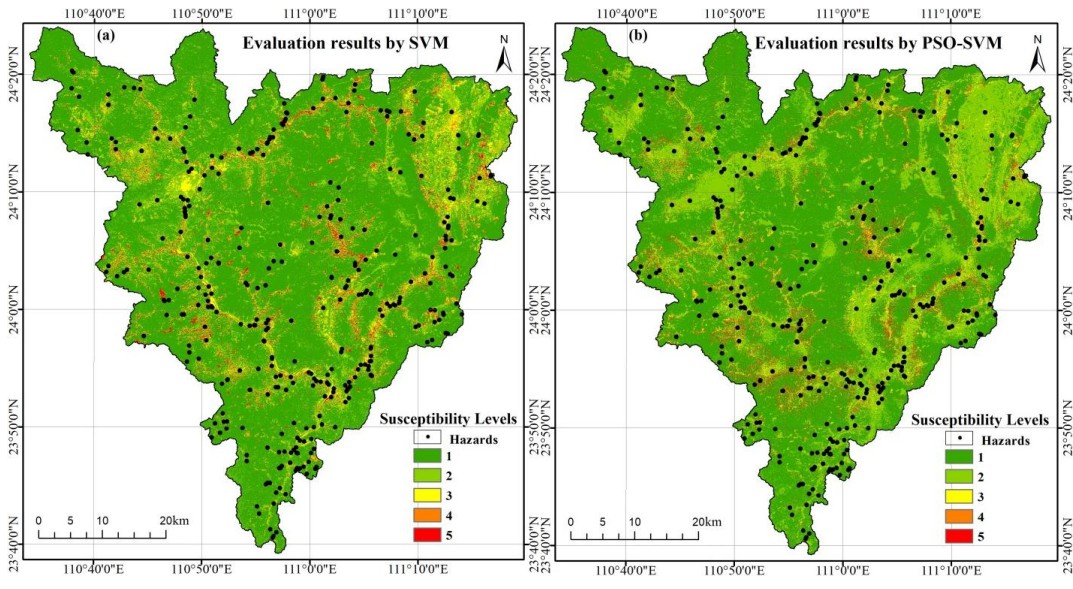
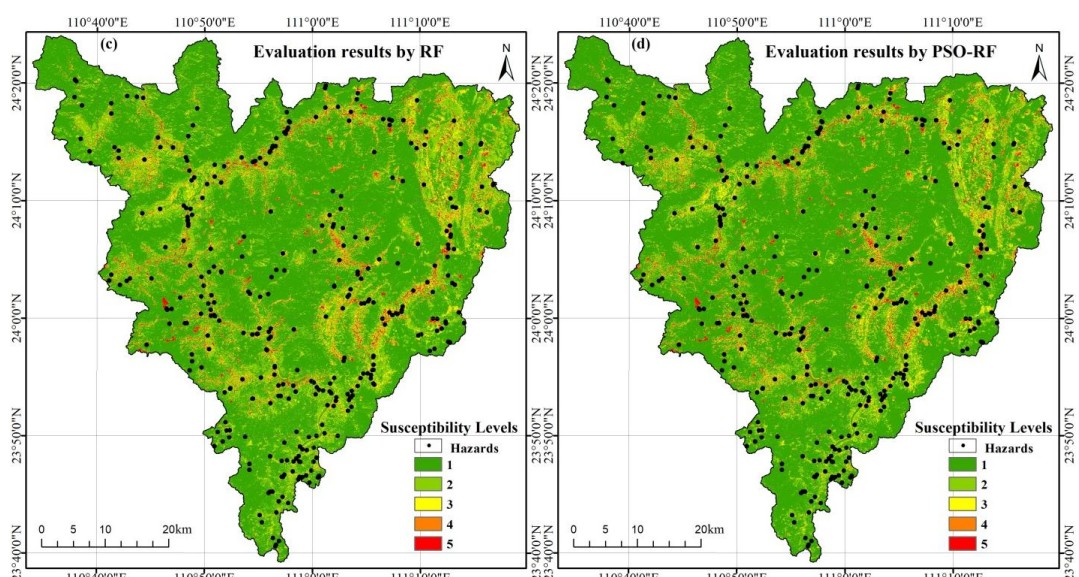

**Fig. 6.** Landslide susceptibility overlying maps of field survey and evaluation results for four ML models in

Zhaoping County [1-extremely low, 2-low, 3-middle, 4-high, 5-extremely high;

(a) SVM, (b) PSO-SVM, (c) RF, (d) PSO-RF]

Figure 6 indicates that the landslide susceptibility evaluation results of four ML models in

Zhaoping County are in accord with the distribution of landslide points of field investigation,

which further illustrates that the methods in evaluating landslide susceptibility in the present

paper was reasonable and effective.

Overall, the ML models of the SVM, PSO-SVM, RF, and PSO-RF achieved excellent

performance in predicting and evaluating the susceptibility levels of landslide in this study.


## 5. Conclusions


The improvement of performance for landslide susceptibility models is still the focus of
widespread concern in the disaster research community, because the capability of the models is
dominated by the method adopted (Tien Bui et al., 2016); though ML methods have been
validated efficient in terms of prediction and assessment performance (Pham et al., 2016).
Therefore, four widely used ML models such as SVM, PSO-SVM, RF, and PSO-RF were
investigated to predict and evaluate the susceptibility levels of landslide for Zhaoping County in
Guangxi of southern China.
Analysis and comparison of the results denoted that all four ML models performed well for
the landslide susceptibility evaluation and prediction as the AUC values of ROC curves are all
greater than 86%. Thereinto, it has been shown that the PSO-RF model (93.4%) has the highest
accuracy in comparison to other landslide models, followed by the PSO-SVM model (91.8%), the
RF model (88.6%), and the SVM model (86.3%). Moreover, the results also showed that the PSO
algorithm has a good effect on SVM and FR models. In addition, our results also revealed that the
PSO-RF and PSO-SVM landslide models have the strong robustness and stable performance, and
those two models are prospective methods that could be applied to landslide susceptibility
evaluation in similar natural geological and ecological environment background regions.
At the same time, the results described in the present study proved that the prediction results
of four ML models are consistent with the field survey results by comparing Fig. 4 and Fig. 6,
which verified the validity of the four ML models again. This also proved that the ML models
have excellent performance in evaluating and predicting the occurrence of landslide. Furthermore,





the results can provide informational service and decision support for landslide early warning,
land use planning and environmental management for local government departments.

In addition, our study found that the10 disaster-related factors selected in this paper can fully

reflect the natural geological and ecological environment characteristics of the study area, and
have a great correlation to the occurrence of landslide disasters. Simultaneously our study also
found that the selection of training samples will affect the susceptibility evaluation results during
the process of landslide susceptibility evaluation using four ML methods. It is worth mentioning
that there is a great difference between the extremely low and extremely high susceptibility
regions for the evaluation results of RF and PSO-RF model, and the occurrences of the extremely
low prone regions is almost 0. However, regions where landslide hazards have not occurred do
not mean that landslide will not occur, so future investigations should pay more attention to
over-fitting in evaluating and predicting the susceptibility of landslide for the RF and PSO-RF
models.



## Code availability


The following program is used to optimize parameters in SVM, PSO-SVM, RF, and PSO-RF,
and further use the optimized parameters to set up training models of SVM, PSO-SVM, RF, and
PSO-RF.
Name of code: gaSVMcgForClass.m, SVMcgForClass.m, main.py
Developer and contact address: Kong Chunfang, Wang Junzuo
Telephone number and E-mail: +8618602766895, kongcf@cug.edu.cn
Year first available: YES
Hardware required:CPU-i5, MEMORY-4G
Software required: WIN10, matlab R2018a, Spyder
Program language: M language, Python
Program size: 9.35k
The code can be accessed using the following link: https://github.com/kongcf/mycode.git



## Data availability

All data used during the study are available in 4TU Research Data repository and can be

accessed through this doi link: https://doi.org/10.4121/12857417.v1





## Author contributions

All the authors made significant contributions to the work. Kong C.F. and Xu K. designed

the research, analyzed the results, and accomplished the validation work; Wang J.Z. completed

the data acquisition, analysis or interpretation; Wu C.L. and Liu G. provided advice for the

revision of the paper. All authors give their final approval of the manuscript version to be

submitted and any revised version of it.



## Competing interests

The authors declare that they have no conflict of interest.



## Acknowledgments

The authors would like to thank the Guangxi Geological Survey Bureau for providing the

various data sets used in this paper. This work has been supported by the National Natural

Science Foundation of China (No: 41201193); Guizhou Science and Technology Planning Project:

Research and Application of Three-dimensional Prediction System for Gold Deposit in Southwest

Guizhou Based on Geological Big Data ([2020]4Y039); Research and Development of Big Data

Management and Intelligent Processing System for Manganese Ore Exploration ([2017]2951);

Open fund project of National-Local Joint Engineering Laboratory on Digital Preservation and

Innovative Technologies for the Culture of Traditional Villages and Towns (CTCZ19K01), and

Open research project of key laboratory of Tectonics and Petroleum Resources (China University

of Geosciences), Ministry of Education (No: TPR-2019-11). The authors would like to thank the

anonymous reviewers for providing valuable comments on the manuscript.

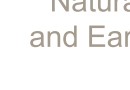

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
