# Peer review of "Landslide susceptibility assessment based on different machine-learning methods in Zhaoping County of eastern Guangxi"

_Natural Hazards and Earth System Sciences, 2020_

## Referee Comment (RC1) · Anonymous Referee #1 · 30 Aug 2020

The current study provided results of landslide risk analyses based on ML algorithms. The writing is generally good without grammar errors. The reviewer would like to provide a few comments as follows:

1. Lines 26-28: Is there any literature research to support such a claim that the variation of robustness and performance of these two models can be neglected among applications in different regions?

2. In the introduction, there is in lack of a summary of the popularity of these ML algorithms. Such a summary can help readers understand why the authors chose these ML algorithms in the current study.

[Figure]

3. The objectives of the introduction should be supported by the gap in the literature. The current objectives jump out from nowhere without any rationale or reasoning.

4. Fig.1: Please highlight the experimental site in mainland China only and ignore the outlying islands to maximize the area of interest.

5. Lines 107-108, 117-120, 128-130: Please provide proper references.

6. Lines 124-127: Why were they chosen? Were there any prior studies to support such a decision?

7. Line 140: Please define "heavy rain".

8. Line 165: Is the LULC map created based on Landsat imagery? Please provide detailed calibration and validation results so the LULC data can be used. LULC determination is not a straightforward process and can be complicated.

9. Fig. 2: For each subplot, please provide numerical ranges for each "grade" based on Table 1.

10. Fig. 3: Please provide more details for each step in the text

11. Tables 2-4: Please merge information of these steps into the text. Tables are used to display arrayed data.

12. Fig. 4: Maybe I missed this – Is the definition of the levels hidden somewhere in the text?

13. Please define robustness in this case – my definition of robustness is that the algorithm consistently delivers good results at all kinds of environments. I don't see how your analyses reflect such quality.

14. I don't understand the message delivered by Table 5. It looks that the accuracy is not good because the landslide points that fell into high susceptibility areas are rare. Please highlight the message delivered by this table.

15. Overall, there is a serious issue with this manuscript. This manuscript simply applied several known algorithms without interpretations. To make it publishable, interpretation of results is required. Why are certain algorithms performing better? Why are certain factors having a higher influence? What does this information mean to management and disaster prevention? These are simply some quick examples on top of my head.

---

## Short Comment (SC1) · 31 Aug 2020

There are some problems we want to discuss with you: 1.What is your research question? Can you really claim in 2020 that the aim of the research was to compare algorithms and their respective performance? How many articles are out there with the exact same question and structure and plots? This field of geomorphology has become an empty shell with no research question whatsoever other than let's measure the delta AUC and let's see how many decimal places down the line, we can claim a model to be better than the other. 2.there are hundreds of articles published every year on model comparison. They are all equally vague and they equally do not provide any practical

solution to a real problem. To prove this statement to you, I would suggest you to search on Scopus using the following keywords: "Landslide susceptibility", "comparison" (and possibly "ensemble"). All the articles will have the same structure, similar results and similar conclusions as those in the present manuscript. To me, this looks more like a technical report rather than a scientific contribution of relevance, sorry. 3.The sampling strategy There is obvious non-parallel data between landslide point and non-landslide point (1:6). How to avoid machine learning preference?
* * *

---

## Referee Comment (RC2) · Anonymous Referee #2 · 7 Oct 2020

**General comments**

Two machine learning methods (SVM, RF) combined with particle swarm optimization (PSO) support, total four models were used to evaluate landslide susceptibility in this paper. The results show that the PSO algorithm has a good improvement on SVM and RF models. This paper has a clear research framework and the outcomes I believe it is beneficial to readers. However several issues are not clearly stated in the current article that need more clarifications. For example, the landslide inventory used in this study is very important since it directly affects the performance of susceptibility models. However the detail information of landslide inventory such as a single rainfall event inventory or a compiled inventory cover different events, material types (rock, earth, soil, mud, debris), landslide patterns (new landslides or old landslides), minimum mapping area, generation methods (satellite or aerial images interpretation, field investigation ) etc. are not mentioned in the article. The second issue, multi-temporal satellite images have been widely used to extract the LULC and NDVI information under different time backgrounds. However, in this paper the only one date (2017/12/24) Landsat 8 OLI image was used to extract temporal information like the LULC and NDVI seeming insufficient to reflect the temporal variation of land covers. The third issue, in order to more completely establish the relation of rainfall scale, natural environmental characteristics, and LULC change with the landslide occurrence potential through the susceptibility model. In my opinion, it's necessary to build the landslide susceptibility model by using the landslide inventories compiled from different scale of rainfall events. For the predisposing factors, except those geology and geomorphological factors, some temporal predisposing factors like rainfall intensity or rainfall accumulation of each event as well as environmental factors like land use, vegetation cover etc. are also suggested to consider. Finally, I can't find the results or discussions for part of the goals and conclusions (P. 5 and P. 29, please see specific comments). Overall speaking, the several main issues mentioned above are suggested to improve before this paper can be considered for publication.

**Specific comments**

1. P. 5, Ln 79-80, "(1) determine the landslide susceptibility assessment factors by multi-source data fusion and correlation factor analysis", What do you mean "data fusion"? And I didn't see "correlation factor analysis" in the article.
2. P. 7, Ln 108, "…there are 345 hidden danger points of landslide…", please explain what "hidden danger points" means.
3. P. 8, Table 1, the classification interval for aspect level 1 is 22.5°, for level 8 is 67.5°, for the remaining six levels the interval is 45°. Why the classification

interval for level 1 and level 8 is different? Additionally, the level 1 and level 8 actually reflect similar aspect, however the extreme different grading number (1 and 8) could seriously affect landslide susceptibility. Please give more explanations.

4. P. 15-16, P. 18-19, the description of basic theory for SVM and RF model can be simplified but need some reference citations.

5. P. 25, Ln 342-344, "…because of the sensitivity of the RF and PSO-RF models to the proportion of landslide samples, it is necessary to carry out sample screening before using RF and PSO-RF models to evaluate the susceptibility of landslide.", this description is unclear. Please explain more about the sensitivity results and how to carry out sample screening?

6. P. 25, Table 5, how to calculate the percentage of landslide points in different susceptibility? The summation of the percentage number in each column should be 100?

7. P. 26, Ln 360, please give more explanation for the field investigation results.

8. P. 29, Ln 398-400, "…our study also found that the selection of training samples will affect the susceptibility evaluation results during the process of landslide susceptibility evaluation using four ML methods.", I can't find the discussion in the manuscript?

***Technical corrections***

1. P. 2,Ln 25, the method "RF" was misspelled as "FR".

2. P. 3,Ln 51, the author name of reference (Tien Bui et al., 2012) was repeated.

---

## Author Comment (AC1) · 17 Oct 2020

**Response to RC1 from Anonymous Referee #1**

October 17, 2020

Dear Reviewer #1:

Thank you for your comments concerning our manuscript ID nhess-2020-251 (Landslide susceptibility assessment based on different machine-learning methods in Zhaoping County of eastern Guangxi). Those comments are all valuable and very helpful for revising and improving our paper, as well as of important guiding significance to our researches. We have studied comments carefully and have made correction which we hope meet the suggestions. Revised portion are marked in highlight in the paper. The main corrections in the paper and the responds to the reviewer's comments are as flowing.

1. **Lines 26-28: Is there any literature research to support such a claim that the variation of robustness and performance of these two models can be neglected among applications in different regions?**

   Zhou et al. (2018) used the ML methods in the landslide susceptibility analysis of Longju in the Three Gorges Reservoir area. Its result showed that the SVM model has a better performance and a strong robustness. Hence, the SVM model

can be recommended before reaching a consensus on the model of landslide susceptibility assessment.

Deng et al. (2018) analyzed the long-term observation test of temperature and gases in the gob of 40106 fully mechanized top-coal caving face at Dafosi coal mine using random forests. At the same time, the particle swarm optimization (PSO) algorithm was employed to optimize the hyper-parameters of RF and SVM for establishing the PSO-RF and PSO-SVM prediction models with optimized parameters. The results indicated that PSO-RF and PSO-SVM models had strong generalization and robustness, and the PSO-RF model could be further applied to other energy and fuel fields.

The relevant literature is as follows:

Zhou, C., Yin, K., Cao, Y., Ahmed, B., Li, Y., Catani, F., and Pourghasemi, H.R.; Landslide susceptibility modeling applying machine learning methods: A case study from Longju in the Three Gorges Reservoir area, China, Comput. Geosci., 2018, 112, 23–37.

Deng, J., Lei, C., Cao, K., Ma, L., Wang, C., and Zhai, X.; Random forest method for predicting coal spontaneous combustion in gob, J. China Coal Soc., 2018, 43(10), 2800–2808.

The above literatures have been labeled in the paper; please see L45, L47-48, L338, and L398.

2. **In the introduction, there is in lack of a summary of the popularity of these ML algorithms. Such a summary can help readers understand why the authors chose these ML algorithms in the current study.**

In the introduction of this paper, examples are given to illustrate that more and more machine learning (ML) algorithms have been optimized and applied for landslide susceptibility assessment in different regions. These have all been

used to quantitatively predict and assess the susceptibility for landslide in different regions of the world. These studies play an important role in the susceptibility evaluation and prediction of landslide. Please see L40-58. At the same time, many comparative studies on landslide susceptibility assessment using different ML methods have been performed. Please see L59-72. These previous studies have shown that the SVM and RF have been widely proved to be useful methods in the evaluation of landslide susceptibility (Marjanović et al., 2011; Tien Bui et al., 2012; Kavzoglu et al., 2014; Trigila et al., 2015; Pham et al., 2016; Ada and San, 2018). However, few studies have focused on the optimization of SVM and RF models in landslide susceptibility prediction and evaluation and compared the optimized results. Therefore, based on previous works, the objective of the present paper is to: (1) optimize SVM and RF models by using a particle swarm optimization (PSO) algorithm; (2) analyze and evaluate the susceptibility levels of landslide by using the SVM, PSO-SVM, RF, and PSO-RF models for Zhaoping County; and (3) compare the performances of four ML models for landslide susceptibility evaluation by receiver operating characteristic (ROC) curve, statistical analysis, and field-verified methods. Please see L74-84.

3. **The objectives of the introduction should be supported by the gap in the literature. The current objectives jump out from nowhere without any rationale or reasoning.**

In the introduction of this paper, L40-56 illustrated that more and more machine learning (ML) algorithms have been optimized and applied for landslide susceptibility assessment in different regions. At the same time, L59-72 illustrated that many comparative studies on landslide susceptibility assessment using different ML methods have been performed. These previous studies have shown that the SVM and RF have been widely proved to be useful methods in the evaluation of landslide susceptibility. However, few studies have focused on the optimization of SVM and RF models in landslide susceptibility prediction and evaluation and

compared the optimized results. Therefore, the objective of the present paper is to: (1) determine the landslide susceptibility assessment factors by multi-source data fusion and correlation factor analysis; (2) optimize SVM and RF models by using a particle swarm optimization (PSO) algorithm; (3) analyze and evaluate the susceptibility levels of landslide by using the SVM, PSO-SVM, RF, and PSO-RF models for Zhaoping County; and (4) compare the performances of four ML models for landslide susceptibility evaluation by receiver operating characteristic (ROC) curve, statistical analysis, and field-verified methods. The results provide valuable informational support for the prediction and evaluation of landslide in Zhaoping County, Guangxi. Please see L40-86.

4. **Fig.1: Please highlight the experimental site in mainland China only and ignore the outlying islands to maximize the area of interest.**

Figure 1 show the location of the study area of Zhaoping County in China and Guangxi Province. The island is an inseparable part of China, so it should not be ignored.

5. **Lines 107-108, 117-120, 128-130: Please provide proper references.**

These materials come from the field investigation report of the geological hazard project by Guangxi Geological Survey Bureau (Huang and He, 2018), and references have been added to the corresponding positions in the article. Thank you for your careful reviews. Please see revised L107-109, L118-121, L125-129, and L130-132.

6. **Lines 124-127: Why were they chosen? Were there any prior studies to support such a decision?**

According to the field investigation report of the geological hazard project by Guangxi Geological Survey Bureau and the disaster factors correlation analysis, a total of ten factors of high correlation with landslide disaster occurrence

were selected as landslide hazard assessment factors for Zhaoping County. Related reference materials have been added in the paper; please see the revised L125-129.

7. **Line 140: Please define "heavy rain".**

Heavy rain generally refers to rainfall with a daily rainfall of 25-49.9 mm (24 hours) or a rainfall of 8.1-16.0 mm per hour.

8. **Line 165: Is the LULC map created based on Landsat imagery? Please provide detailed calibration and validation results so the LULC data can be used. LULC determination is not a straightforward process and can be complicated.**

The LULC map in the paper comes from the manual visual interpretation results of Landsat 8 OLI image (2017/12/24, 124/043). The interpretation results are shown in Figure 2(h).

The process of interpretation is as follows: (1) Radiometric calibration and atmospheric correction for the Landsat 8 OLI image; (2) Based on the ground control point, geometric correction for the Landsat 8 OLI image, and the correction accuracy is less than 1 pixel; (3) Combined with topographic map and China's Land Use/Cover Dataset (CLUD) of 2015 by Institute of Geographic Sciences and Natural Resources Research, Chinese Academy of Sciences, the LULC of Zhaoping County was divided into five categories by manual visual interpretation: 1-cultivated land; 2-woodland; 3-grassland; 4-river and lake; 5-construction land; (4) By randomly selecting 10% samples to verify the accuracy of field investigation, it was shown that the overall classification accuracy is more than 95.53%, which meets the accuracy requirements of this paper.

The LULC map is only a representative factor reflecting the impact of human activities on the environment in Zhaoping County, and due to the limited space

of the paper, detailed calibration and verification results are not provided in the paper.

9. **Fig. 2: For each subplot, please provide numerical ranges for each "grade" based on Table 1.**

The level of each subplot in Figure 2 is one-to-one corresponding to the specific numerical range in Table 1.

For example, as shown in Figure 2(a) of the slope, 1 represents 0-7°, 2 represents 7-13°, 3 represents 13-25°, 4 represents 19-25°, 5 represents 25-34°, 6 represents 34-50°, 7 represents 50-70°, and 8 represents 70-76°.

Figure 2(b) of the aspect, 1 represents 337.5-22.5°, 2 represents 22.5-67.5°, 3 represents 67.5-112.5°, 4 represents 112.5-157.5°, 5 represents 157.5-202.5°, 6 represents 202.5-247.5°, 7 represents 247.5-292.5°, and 8 represents 292.5-337.5°.

Figure 2(c) of the plan curvature, 1 represents -25–5°, 2 represents -5–2.5°, 3 represents -2.5–1°, 4 represents -1-0°, 5 represents 0-1°, 6 represents 1-2.5°, 7 represents 2.5-5°, and 8 represents 5-28.9°.

Figure 2(d) of the annual rainfall, 1 represents 0-1980 mm, 2 represents 1980-2100 mm, 3 represents 2100-2220 mm, 4 represents 2220-2340 mm, 5 represents 2340-2460 mm, 6 represents 2460-2580 mm, 7 represents 2580-2700 mm, and 8 represents 2700-2820 mm.

Figure 2(e) of the NDVI, 1 represents 0-0.01, 2 represents 0.01-0.09, 3 represents 0.09-0.17, 4 represents 0.17-0.25, 5 represents 0.25-0.33, 6 represents 0.33-0.4, 7 represents 0.4-0.5, and 8 represents 0.5-0.71.

Figure 2(f) of the stratum lithology, 0 represents river, 1 represents Quaternary, 2 represents carbonate rock, 5 represents clasolite intercalated with siliceous rocks, 6 represents clastic rock, 7 represents sandstone and shale, and 8 represents granite or basal rocks.

Figure 2(g) of the tectonic complexity, 1 represents 0-1.4, 2 represents1.4-2.7, 3 represents 2.7-3.8, 4 represents 3.8-4.9, 5 represents 4.9-6, 6 represents 6-7.3, 7 represents 7.3-8.9, and 8 represents 8.9-9.4.

Figure 2(h) of the LULC, 1 represents cultivated land, 2 represents woodland, 3 represents grassland, 4 represents river and lake, and 5 represents construction land.

Figure 2(i) of the residential density, 1 represents 0-1.2, 2 represents1.2-2.7, 3 represents 2.7-4.5, 4 represents 4.5-6.9, 5 represents 6.9-10.1, 6 represents 10.1-14.2, 7 represents 14.2-19.7, and 8 represents 19.7-25.

Figure 2(j) of the road network density, 1 represents 0-3.2, 2 represents3.2-4.7, 3 represents 4.7-6.1, 4 represents 6.1-7.8, 5 represents 7.8-9.7, 6 represents 9.7-11.7, 7 represents 11.7-13.9, and 8 represents 13.9-14.

10. **Fig. 3: Please provide more details for each step in the text**

Figure 3 is a Flowchart of landslide susceptibility evaluation based on ML. For details, please refer to the paper of 3.1-3.4.

11. **Tables 2-4: Please merge information of these steps into the text. Tables are used to display arrayed data.**

Tables 2-4 are the specific steps of the four ML algorithms, so, it is appropriate to put it in a table rather than merge it into the text.

12. **Fig. 4: Maybe I missed this – Is the definition of the levels hidden some-where in the text?**

Figure 4 is the evaluation results of landslide susceptibility for four ML models in Zhaoping County, and 1 represents extremely low susceptibility, 2 represents low susceptibility, 3 represents middle susceptibility, 4 represents high susceptibility, 5 represents extremely high susceptibility.
13. **Please define robustness in this case – my definition of robustness is that the algorithm consistently delivers good results at all kinds of environments. I don't see how your analyses reflect such quality.**

    Robustness in this paper refers to the stable performance of the PSO-RF and PSO-SVM models established in this paper, that is, by inputting the attribute values of each evaluation factor, the results of the susceptibility level of landslide disasters in the study area are obtained, and the results are in good agreement with the results of field investigation.

    The reason why the PSO-RF and PSO-SVM models have strong robustness is that we applied the models in this paper to evaluate the landslide susceptibility in 23 other regions of Guangxi and the evaluation results agree with the results of field investigation.

14. **I don't understand the message delivered by Table 5. It looks that the accuracy is not good because the landslide points that fell into high susceptibility areas are rare. Please highlight the message delivered by this table.**

    Table 5 indicates the proportions of hazards points falling into different susceptibility levels.

    The first line indicates that the PSO-RF model simulates the probability of the landslide point falls into the extremely high susceptibility level is the highest, which is 0.2306%, followed by the PSO-SVM model, the third is the RF model, and the last one is the SVM model.

    The second line indicates that the PSO-RF model simulates the probability of the landslide point falls into the high susceptibility level is also the highest, which is 0.0845%, followed by the PSO-SVM model, the third is the RF model, and the last one is the SVM model.

    The third line indicates that the PSO-RF model simulates the probability of the landslide point falls into the middle susceptibility level is the lowest, which is

0.0117%, followed by the RF model, the third is the PSO-SVM RF model, and the last one is the SVM model.

The fourth line indicates that the PSO-RF model simulates the probability of the landslide point falls into the low susceptibility level is also the lowest, which is 0.0041%, followed by the PSO-SVM model, the third is the RF model, and the last one is the SVM model.

The fifth line indicates that the PSO-RF model simulates the probability of the landslide point falls into the extremely low susceptibility level is also the lowest, which is 0.0005%, followed by the PSO-SVM model, the third is the RF model, and the last one is the SVM model.

The above analysis shows that the proportion of landslide disaster points simulated by the PSO-RF model falling into extremely high and high-prone areas is higher than that of other models. At the same time, the proportion of landslide disaster points simulated by the PSO-RF model falling into low and extremely low-prone areas is lower than that of other models, which from another aspect shows that the PSO-RF model has the highest simulation accuracy and the best performance in comparison to other landslide models.

15. **Overall, there is a serious issue with this manuscript. This manuscript simply applied several known algorithms without interpretations. To make it publishable, interpretation of results is required. Why are certain algorithms performing better? Why are certain factors having a higher influence? What does this information mean to management and disaster prevention? These are simply some quick examples on top of my head.**

It has been discussed in the introduction that this paper is based on previous works, and many previous studies have proved that SVM and RF models have better performance in landslide susceptibility evaluation and prediction than other models. Based on this, the focus of the present paper is to optimize SVM and

RF models by using a particle swarm optimization (PSO) algorithm; analyze and evaluate the susceptibility levels of landslide by using the SVM, PSO-SVM, RF, and PSO-RF models for Zhaoping County; and compare the performances of four ML models for landslide susceptibility.

Our research denoted that the PSO algorithm has a good effect on SVM and RF models. Meanwhile, our research also demonstrated that PSO-RF model has a better prediction performance than the PSO-SVM model, which is mainly due to the large number of factors selected in this study, the PSO-RF model, a type of ensemble learning, exhibited advantages over a traditional ML method by not only accounting for different types of factors but also evaluating the relative importance of the factors in terms of landslide stability. The relevant discussion has been added to the paper, Please see L341-346.

Our research also denoted that the simulation results of the paper proved that the occurrence of landslide disasters has a strong correlation with the stratum lithology, geological tectonic complexity, precipitation, human engineering activities, and vegetation index. This is mainly due to the occurrence of landslide disasters have its internal and external factors: stratum lithology, geological tectonic complexities are the internal causes of landslide disasters; precipitation, human engineering activities, and vegetation cover are the external causes of landslide disasters. Internal causes play a major role in the occurrence of landslide, while external causes play a role in promoting the occurrence of landslide.

To sum up, the information of the landslide susceptibility levels from the results can provide method support for engineering construction, ecological environment construction, rapid economic development, disaster reduction and disaster prevention in Zhaoping County, Guangxi.

Once again, we are very grateful for your comments, and those comments are all valuable and very helpful for revising and improving our paper, as well as of important
guiding significance to our researches.

---

## Author Comment (AC2) · 17 Oct 2020

**Response to SC1 from Zhu Liang**

October 17, 2020

Dear Zhu Liang:

Thank you for your comments concerning our manuscript ID nhess-2020-251 (Landslide susceptibility assessment based on different machine-learning methods in Zhaoping County of eastern Guangxi). Those comments are all valuable and very helpful for revising and improving our paper, as well as of important guiding significance to our researches. We have studied comments carefully and have made correction which we hope meet the suggestions. Revised portion are marked in highlight in the paper. The main corrections in the paper and the responds to the reviewer's comments are as flowing:

1. **What is your research question? Can you really claim in 2020 that the aim of the research was to compare algorithms and their respective performance? How many articles are out there with the exact same question and structure and plots? This field of geomorphology has become an empty shell with no research question whatsoever other than let's measure the delta AUC and let's see how many decimal places down the line, we can claim a model to be better than the other.**

The geological environment in eastern Guangxi is fragile and geological hazards occur frequently, which not only causes huge economic losses and ecological environment damage, but also seriously restricts the survival of human beings and the sustainable development of human society (Pourghasemi et al., 2012; Uitto and Shaw, 2016). In particular, with the rapid development of the economy in recent decades, the frequency and intensity of the geological hazards are rapidly increasing with the over-exploitation and utilization of natural resources by humans (Guzzetti et al., 1999). Therefore, it is of great significance to objectively evaluate the susceptibility of geological hazards for the reduction and prevention of the disasters. Please see L32-39.

The objective of the present paper is to seek a method to quickly and accurately evaluate the susceptibility grade of landslide for Zhaoping County of Guangxi. To this end, a total of ten factors of high correlation with landslide disaster occurrence were selected for running four machine-learning (ML) methods, and the landslide susceptibility grade was quickly and accurately evaluated by model simulation in Zhaoping County, providing methodological support for engineering construction, ecological environment construction, rapid economic development, and disaster reduction and disaster prevention for Guangxi.

2. **There are hundreds of articles published every year on model comparison. They are all equally vague and they equally do not provide any practical solution to a real problem. To prove this statement to you, I would suggest you to search on Scopus using the following keywords: "Landslide susceptibility", "comparison" (and possibly "ensemble"). All the articles will have the same structure, similar results and similar conclusions as those in the present manuscript. To me, this looks more like a technical report rather than a scientific contribution of relevance, sorry.**

Indeed, I admit that hundreds of articles published every year on model comparison, and each researcher wants to explore a common model for rapid and

accurate evaluation the susceptibility grade for landslide disasters. However, this work is more complex and requires long-term efforts by all researchers.

At the same time, the occurrence of landslide disasters is very different in different regions and different geological environment, so, a large number of landslide models are produced for different regions. This paper aims at the ecological environment characteristics of landslide disaster in Zhaoping County, hoping to find a suitable model to quickly and accurately evaluate the susceptibility grade for landslide disaster in Guangxi.

3. **The sampling strategy. There is obvious non-parallel data between landslide point and non-landslide point (1:6). How to avoid machine learning preference?**

Field investigation showed that there were 345 landslide disaster sites in the study area, all of which participated in the training and testing of the model. Among them, 242 (70%) landslide hazards points were selected as training set samples, and 103 (30%) landslide disaster points were selected as testing set samples. For non-disaster points, training and testing datasets were constructed by random sampling method based on environmental similarity. Among them, 1,251 non-hazards points with low environmental similarity with landslide disaster points were selected as training set samples, and 939 non-hazards points with low environmental similarity with landslide disaster points were selected as testing set samples. Therefore, in view of the obvious non-parallel data between landslide points and non-slide points in the study area, random sampling method based on environmental similarity strategies was adopted to construct training set and testing set to avoid machine learning preference. Please see the L181-182 and L183-184.

In summary, we are very grateful for your comments, and those comments are all valuable and very helpful for revising and improving our paper, as well as of important

guiding significance to our researches.

---

## Author Comment (AC3) · 17 Oct 2020

**Response to RC2 from Anonymous Referee #2**

October 17, 2020

Dear Anonymous Referee #2:

Thank you for your comments concerning our manuscript ID nhess-2020-251 (Landslide susceptibility assessment based on different machine-learning methods in Zhaoping County of eastern Guangxi). Those comments are all valuable and very helpful for revising and improving our paper, as well as of important guiding significance to our researches. We have studied comments carefully and have made correction which we hope meet the suggestions. Revised portion are marked in highlight in the paper. The main corrections in the paper and the responds to the reviewer's comments are as flowing:

**1 General comments**

1. **Two machine learning methods (SVM, RF) combined with particle swarm optimization (PSO) support, total four models were used to evaluate landslide susceptibility in this paper. The results show that the PSO algorithm has a good improvement on SVM and RF models. This paper has a clear**

**research framework and the outcomes I believe it is beneficial to readers. However several issues are not clearly stated in the current article that need more clarifications.**

We appreciate for reviewers' approval, and have same opinions with the reviewer. Once again, thank you very much for your approval.

2. **For example, the landslide inventory used in this study is very important since it directly affects the performance of susceptibility models. However the detail information of landslide inventory such as a single rainfall event inventory or a compiled inventory cover different events, material types (rock, earth, soil, mud, debris), landslide patterns (new landslides or old landslides), minimum mapping area, generation methods (satellite or aerial images interpretation, field investigation) etc. are not mentioned in the article.**

The landslide inventory map in Zhaoping County was prepared from field investigation of Guangxi Geological Survey Bureau (Huang and He, 2018). Please see L120-121. And according to the field investigation report of the geological hazard project by Guangxi Geological Survey Bureau in 2018, there are 345 landslide disaster points in Zhaoping County (Huang and He, 2018). Please see L107-109. These landslide disaster points are used as landslide disaster training and testing samples without considering landslide properties, material types, patterns, minimum mapping area, and generation methods and so on.

3. **The second issue, multi-temporal satellite images have been widely used to extract the LULC and NDVI information under different time backgrounds. However, in this paper the only one date (2017/12/24) Landsat 8 OLI image was used to extract temporal information like the LULC and NDVI seeming insufficient to reflect the temporal variation of land covers.**

We have same opinions with the reviewer. One date (2017/12/24) Landsat 8 OLI

image, which is close to the field investigation time, was selected to extract the LULC and NDVI index for the Zhaoping County.

When the LULC and NDVI index, as landslide related factors, are put into four ML models with the other eight factors, they only reflects the status of the study area and do not need to reflect the temporal variation. So one date Landsat 8 OLI image on is enough.

4. **The third issue, in order to more completely establish the relation of rainfall scale, natural environmental characteristics, and LULC change with the landslide occurrence potential through the susceptibility model. In my opinion, it's necessary to build the landslide susceptibility model by using the landslide inventories compiled from different scale of rainfall events. For the predisposing factors, except those geology and geomorphological factors, some temporal predisposing factors like rainfall intensity or rainfall accumulation of each event as well as environmental factors like land use, vegetation cover etc. are also suggested to consider.**

A total of ten factors of high correlation with landslide disaster occurrence were chosen based on the field investigation report of the geological hazard project by Guangxi Geological Survey Bureau and the disaster factors correlation analysis in Zhaoping County: slope, aspect, curvature, annual rainfall, NDVI, stratum lithology, tectonic complexity, LULC, residential density, and road network density County (Huang and He, 2018). These factors reflect the geological environment characteristics for Zhaoping County, including geological characteristics, geomorphological characteristics, meteorological characteristics, ecological characteristics, environmental characteristics, and human activities characteristics, and so on. Please see L125-129.

5. **Finally, I can't find the results or discussions for part of the goals and conclusions (P5 and P29, please see specific comments).**

The questions of P5 and P29 have been discussed in part of specific comments. Please see comments 1 and 8.

6. **Overall speaking, the several main issues mentioned above are suggested to improve before this paper can be considered for publication.**

We are very grateful for your comments, and those comments are all valuable and very helpful for revising and improving our paper, as well as of important guiding significance to our researches.

**2 Specific comments**

1. **P5, Ln79-80, "(1) determine the landslide susceptibility assessment factors by multi-source data fusion and correlation factor analysis", what do you mean "data fusion"? And I didn't see "correlation factor analysis" in the article.**

"Data fusion" here refers to a series of data processing and analysis, and the main processes are as follows: Firstly, the collected multi-source data are preprocessed, including data screening and correlation factor analysis, in order to determine the landslide disaster assessment factors. Then, the selected landslide disaster assessment factors are standardized and classified. Finally, a standardized data set is obtained to run four ML models.

"Correlation factor analysis" here refers to the analysis process of distinguishing the factors that have high correlation with the occurrence of landslide disaster from all factors provided by field investigation of Guangxi Geological Survey Bureau.

Due to space limit, it is not carefully explained here.
2. **P7, Ln108, "...there are 345 hidden danger points of landslide...", please explain what "hidden danger points" means.**

I am sorry. "Hidden danger points of landslide" should be replaced by "landslide disaster points", it has been corrected in the paper, thank you for your careful reviews, and please see the revised L108.

3. **P. 8, Table 1, the classification interval for aspect level 1 is 22.5°, for level 8 is 67.5°, for the remaining six levels the interval is 45°. Why the classification interval for level 1 and level 8 is different? Additionally, the level 1 and level 8 actually reflect similar aspect, however the extreme different grading number (1 and 8) could seriously affect landslide susceptibility. Please give more explanations.**

I am sorry. Table 1 for aspect level 1 is "[0, 22.5)" should be replaced by "[337.5, 22.5)"; and the level 8 is "[292.5, 360)" should be replaced by "[292.5, 337.5)". It has been corrected in the paper, thank you for your careful reviews, and please see the revised Table 1.

4. **P15-16, P18-19, the description of basic theory for SVM and RF model can be simplified but need some reference citations.**

The reference citations have been added in the paper, and please see the revised L210, L215-216, and L228 of P15-16, and L262, L267-269 of P18.

5. **P25, Ln 342-344, "...because of the sensitivity of the RF and PSO-RF models to the proportion of landslide samples, it is necessary to carry out sample screening before using RF and PSO-RF models to evaluate the susceptibility of landslide.", this description is unclear. Please explain more about the sensitivity results and how to carry out sample screening?**

Our study indicated that the accuracy of landslide disaster prediction is low if the RF and PSO-RF models are used directly, but the prediction accuracy will be

greatly improved, if the RF and PSO-RF models are used for landslide samples selection and then the models prediction are carried out. Therefore, it is necessary to carry out sample screening before using RF and PSO-RF models to evaluate the susceptibility of landslide.

6. **P25, Table 5, how to calculate the percentage of landslide points in different susceptibility? The summation of the percentage number in each column should be 100?**

   Table 5 indicates the percentages of landslide points falling into different susceptibility levels, calculated by dividing the grid number of disaster points falling into different susceptibility levels by the total grid number of this grade, and the total number of this percentage is not 100.

7. **P26, Ln 360, please give more explanation for the field investigation results.**

   Field investigation refers to the onsite investigation carried out by the staff of Guangxi Geological Survey Bureau in 2018, with the purposes of obtaining the landslide inventory map of Zhaoping County (Huang and He, 2018). This has been explained in the 2.1 and 2.2. Please see L107-109 and L120-121.

8. **P29, Ln 398-400, "…our study also found that the selection of training samples will affect the susceptibility evaluation results during the process of landslide susceptibility evaluation using four ML methods.", I can't find the discussion in the manuscript?**

   Our study indicated that the accuracy of landslide disaster prediction is low if the RF and PSO-RF models are used directly, but the prediction accuracy will be greatly improved, if the RF and PSO-RF models are used for landslide samples selection and then the models prediction are carried out. Therefore, it is necessary to carry out sample screening before using RF and PSO-RF models to evaluate the susceptibility of landslide. Please see L353-355. At the same time,

The results also demonstrated that PSO-RF model has a better prediction performance than the PSO-SVM model, which is mainly due to the large number of factors selected in this study, the PSO-RF model, a type of ensemble learning, exhibited advantages over a traditional ML method by not only accounting for different types of factors but also evaluating the relative importance of the factors in terms of landslide stability (Zhang et al., 2017). Please see L341-346.

The above discussions show that the selection of training samples will affect the susceptibility evaluation results in the process of landslide susceptibility evaluation using four ML methods.

**3   Technical corrections**

1. **P2, Ln 25, the method "RF" was misspelled as "FR".**

   The "FR" has been replaced by "RF" and thanks for your careful reviews. Please see revised L26.

2. **P3, Ln 51, the author name of reference (Tien Bui et al., 2012) was repeated.**

   The author name of reference (Tien Bui et al., 2012) has been revised, and thanks for your careful reviews. Please see revised L51.

All editorial errors have been corrected accordingly as suggested by the reviewer throughout the manuscript, and we would appreciate the reviewer for those comments.

In summary, we are very grateful for your comments, and those comments are all valuable and very helpful for revising and improving our paper, as well as of important guiding significance to our researches.